# Learning Semantic Proxies from Visual Prompts for Parameter-Efficient Fine-Tuning in Deep Metric Learning

**Li Ren**[1]**, Chen Chen**[1,2]**, Liqiang Wang**[1]**, Kien Hua**[1]
[1] Department of Computer Science     [2] Center for Research in Computer Vision
University of Central Florida, USA
`{Li.Ren, Chen.Chen, Liqiang.Wang, Kien.Hua}@ucf.edu`

## Abstract

Deep Metric Learning (DML) has long attracted the attention of the machine learning community as a key objective. Existing solutions concentrate on fine-tuning the pre-trained models on conventional image datasets. As a result of the success of recent pre-trained models trained from larger-scale datasets, it is challenging to adapt the model to the DML tasks in the local data domain while retaining the previously gained knowledge. In this paper, we investigate *parameter-efficient methods* for fine-tuning the pre-trained model for DML tasks. In particular, we propose a novel and effective framework based on learning Visual Prompts (VPT) in the pre-trained Vision Transformers (ViT). Based on the conventional proxy-based DML paradigm, we augment the proxy by incorporating the semantic information from the input image and the ViT, in which we optimize the visual prompts for each class. We demonstrate that our new approximations with semantic information are superior to representative capabilities, thereby improving metric learning performance. We conduct extensive experiments to demonstrate that our proposed framework is effective and efficient by evaluating popular DML benchmarks. In particular, we demonstrate that our fine-tuning method achieves comparable or even better performance than recent state-of-the-art full fine-tuning works of DML while tuning only a small percentage of total parameters. Code is available at `https://github.com/Noahsark/ParameterEfficient-DML`.

## 1 Introduction

Metric learning is a crucial part of machine learning that creates distance functions based on the semantic similarity of the data points. Modern Deep Metric Learning (DML) uses deep neural networks to map data to an embedding space where similar data are closer. This is especially useful for computer vision applications, including image retrieval (Lee et al., 2008; Yang et al., 2018), human re-identification (Wojke & Bewley, 2018; Hermans et al., 2017), and image localization (Lu et al., 2015; Ge et al., 2020). Because of the success of the development of deep neural networks, most current DML works (Song et al., 2016; Wang et al., 2017b; Kim et al., 2020; Roth et al., 2022; Yang et al., 2022) recommend *convolutional neural networks* (CNN) (He et al., 2016; Wang et al., 2019) pre-trained on conventional ImageNet1K dataset (Deng et al., 2009) as their backbone of data encoder. It was recently discovered that the more advanced *vision transformers* (ViT) are better in terms of representation capability and performance on DML tasks (El-Nouby et al., 2021; Ramzi et al., 2021; Patel et al., 2022; Ermolov et al., 2022).

The recent foundation models have trends to scale up their parameters and training data. Utilizing larger backbone models and massive amounts of raw data, recent studies have demonstrated superior performance in major deep learning tasks (Radford et al., 2021). Considering their prohibitively expensive training costs, applying these huge models to downstream tasks becomes a crucial and challenging research topic. Unfortunately, the most straightforward adaptation strategy, the *full fine-tuning* of the pre-trained model, has been criticized for its high training cost for tuning all parameters and the catastrophic forgetting caused by the difference between the scale and distribution of the raw data and the local dataset.

Recent research looks into efficient techniques to adapt large-scale models by fine-tuning just a few parameters. A simple method for this is to optimize task-specific heads, such as the linear classifier, commonly known as *linear probing* (Mahajan et al., 2018; Chen et al., 2021). People also investigated more efficient ways to adjust the Transformers, such as adjusting the bias factor (Zaken et al., 2021), tuning soft prompts (Lester et al., 2021; Jia et al., 2022) to serve as additional input data, or tuning additional parameters besides the Transformers (Rebuffi et al., 2018; Hu et al., 2021; Chen et al., 2022a). Recent works also confirm that the ViT can also be effectively adapted with those methods to downstream tasks in both image and video domains (Jia et al., 2022; Chen et al., 2023b; Yang et al., 2023).

However, the parameter-efficient method for fine-tuning a large model, particularly for DML tasks, has yet to be fully investigated. Recent works that propose full fine-tuning of the ViTs in the DML paradigm continue to require a substantial amount of computational resources. For example, the state-of-the-art method proposes training their model on multiple GPUs or a huge batch size for an extended period (Ermolov et al., 2022; Patel et al., 2022). In addition, it is still being determined whether the full fine-tuning is optimized for DML tasks, as inappropriate fine-tuning would easily lead to overfitting, resulting in catastrophic forgetting where previously learned tasks and presentations are lost. To overcome these difficulties, we investigate parameter-efficient methods to fine-tune the pre-trained vision models on DML tasks in this paper. Specifically, we compare existing parameter-efficient fine-tuning strategies based on proxy-based DML paradigms. After that, we chose Visual-Prompt Tuning (VPT), the most efficient strategy, as our backbone to increase its efficiency on DML tasks further.

To this end, we propose an effective learning framework based on the VPT for fine-tuning pre-trained ViTs on DML tasks. Based on the traditional proxy-based DML system, we learn the representation of image samples, particularly by fine-tuning additional learnable soft prompts while other parameters are fixed. Instead of the original proxies that are randomly initialized, we propose to initial proxies with semantic information by training a set of extra visual prompts for each image class. Then, the proxies are generated from the ViT with the class-based visual prompts in each class. Furthermore, we propose a novel method to integrate semantic embeddings in the same class progressively with fresh input samples by feeding them into a *recurrent neural network* efficiently. We show that our technique outperforms the original proxy-based loss regarding learning efficiency and metric learning performance. We further show that by tuning a small proportion of parameters, our framework performs comparably to the state-of-the-art full fine-tuning techniques with a substantially reduced batch size and computing usage. Our main contributions to this paper can be summarized as follows:

1. We investigate and compare several parameter-efficient fine-tuning strategies on pre-trained ViTs for DML tasks using conventional proxy-based DML losses.

2. We propose a simple but effective approach that uses VPT to generate and integrate semantic proxies from image data to improve their representational capacity and DML performance.

3. Extensive experiments on popular DML benchmarks show that our proposed parameter-efficient framework outperforms previous fully fine-tuning state-of-the-art while learning a small percentage of parameters (e.g., 5.2% of 30M).

## 2 RELATED WORKS

**Parameter-Efficient Fine-Tuning.** Firstly introduced by Vaswani et al. (2017), Transformers have been widely applied in NLP and CV areas. Recently, researchers have begun to develop parameter-efficient methods to enhance and accelerate the training process of Transformers. Houlsby et al. (2019) first propose fine-tuning a group of extra-lightweight models as adapters in addition to the backbones. Pfeiffer et al. (2020) designed the adapters with the extra structure that combines down and up projection networks. Jia et al. (2022) demonstrate that the prompts also perform well on ViT for vision tasks. Gao et al. (2023) further extends the prompt tuning in the video recognition area. Khattak et al. (2022) utilize the prompt tuning between the image and language domain, and Chen et al. (2023a) further propose the optimal transport solution to match the prompts between the two domains. Unlike the above works, we propose applying prompt tuning to adapt the pre-trained model specifically for DML tasks and generate proxies containing semantic information to enhance its tuning performance further.

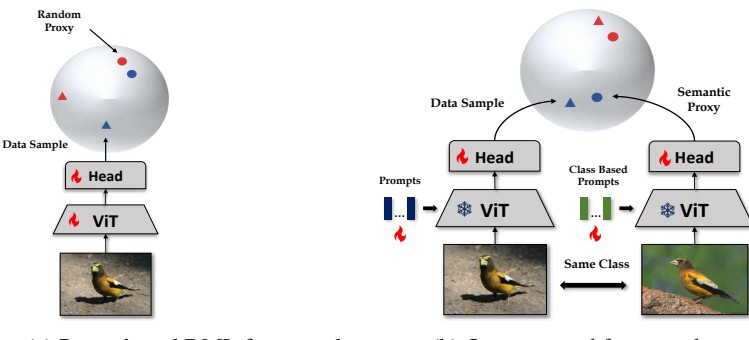

(a) Proxy-based DML framework       (b) Our proposed framework

Figure 1: Overview of a conventional proxy-based DML framework and our proposed framework in the training stage. (a) In a typical proxy-based DML framework with the proxies that are randomly initialized, the ViT encoder is fully fine-tuned. (b) In our proposed framework, we only fine-tune the linear head and additional learnable visual prompts. We also propose to generate proxies that contain semantic information by tuning class-based prompts. The *fire* label represents the learnable parameters, while the *snow* label represents the fixed parameters.

**Deep Metric Learning.** Deep Metric Learning seeks to develop representations and metrics for determining the separation between two data samples. The *contrastive loss* is applied to samples from various classes (Chopra et al., 2005; Hadsell et al., 2006). Another group of methods chooses an additional sample as an anchor to compare with both positive and negative samples using the *triplet loss* (Weinberger & Saul, 2009; Cheng et al., 2016; Hermans et al., 2017). More recently, ProxyNCA (Movshovitz-Attias et al., 2017) utilized *proxies*, a collection of learnable representations, to represent the data classes. Teh et al. (2020) further enhances the ProxyNCA by scaling the proxy gradient. Later, Proxy-Anchor (PA) (Kim et al., 2020) improves the proxy methods by setting the proxies as anchors rather than samples to learn the inter-class structure. In this paper, we adopt the PA as the backbone loss function of our framework for its high performance and reliability. Recent studies also investigated the application of domain adaptation in image or textual retrieval tasks (Laradji & Babanezhad, 2020; Ren et al., 2019; 2021) Wang et al. (2017a) employ domain adaptation to align image and textural data using a single discriminator, whereas Ren et al. (2024) utilize the adversarial learning to align the distributions between data and proxies.

**Deep Metric Learning with Transformer.** People have recently begun to employ pre-trained ViT to improve DML performance because of their greater data representation capacity and performance in CV tasks. El-Nouby et al. (2021) initially proposed ViT on image retrieval tasks with conventional constructive loss. Ermolov et al. (2022) propose comparing the data embeddings from ViT in hyperbolic space, whereas Patel et al. (2022) propose to utilize exceptionally large batch sizes to enhance the DML performance. More recently, Kim et al. (2023) proposes to build a hierarchical structure to enhance the DML performance on hyperbolic space. Wang et al. (2023) factorizes the training signal for different components in the backbone network, and Kotovenko et al. (2023) constructs conditional hierarchical structure with cross attention in the training process. Considering the prompts, Chen et al. (2022b) apply prompts to NLP Transformers in DML paradigms, but they apply fixed prompt terms to enhance full fine-tuning on an NLP task. In contrast, we propose optimizing soft visual prompts for ViTs based on proxy-based DML paradigms in image retrieval tasks, distinguishing us from the previous approaches.

## 3  PRELIMINARY

### 3.1  VISION TRANSFORMERS AND FINE-TUNING

Dosovitskiy et al. (2020) suggested Vision Transformers (ViT) as an extension of the original NLP Transformers (Vaswani et al., 2017). A standard ViT comprises an embedding layer and numerous successively connected blocks. The embedding layer divided an input image sample $I \in \mathbb{R}^{H \times W \times 3}$ into $N$ flattened 2D patches $x$ of the size $k \times k$. The patches are then transmitted into the attention blocks along with additional position embeddings. Like NLP Transformers, the embeddings combine

with an extra learnable [CLS] token $x_{cls}$ as the embeddings for downstream tasks. They are then processed within the transformer blocks, built with a multi-head self-attention (MHSA) architecture. Those embeddings are linearly projected into three vectors in MHSA: query $Q$, key $K$, and value $V$. The self-attention $\mathrm{softmax}(\frac{QK^T}{\sqrt{d}}V)$ paired with an extra LayerNorm and an MLP layer is then used to construct the representation for the following attention layer. After transforming within several transformer blocks, the information from all other patches is passed to a linear head before being sent to the loss of downstream tasks. The size of a typical transformer is determined by the number of block layers $L$ and the dimension $D$ of the hidden embedding within the Transformer.

**Adapter Fine Tuning** (Rebuffi et al., 2018) is one major group of existing works that fine-tune a group of MLP models besides the Transformer. A typical vision transformer adapter (Chen et al., 2022a) consists of a series of linear layers that project the embeddings of dimension $D$ into a low-dimension space of dimension $d$ where $d \ll D$ known as the bottleneck, which is linked to another linear layer that projects up to the original dimension after passing a no-linear unit. An extra residual link connects the starting and finishing locations of this MLP configuration.

**Bitfit Fine Tuning** (Zaken et al., 2021) is one of the simplest solutions to adapt the pre-trained model to a downstream task. Except for the Bias factor of each linear projector, every other parameter is fixed. Since it is a simple plug-and-play solution to integrate with other strategies, we incorporate Bitfit into our framework to enhance its performance.

**Visual Prompt Tuning** (Jia et al., 2022) relies on additional learnable prompt vectors as the extra data input. Inspired by the prompts in the language model, visual prompts combine the original visual embeddings with additional $n$ learnable prompts $T = [t_1, \ldots, t_n] \in \mathbb{R}^{n \times D}$. For the embedding of the image patches $x^k$ in the $k$th layer, the output of this layer and the next layer can be described as follows:

$$[x_{cls}, x_1^{k+1}, \ldots, [\quad]] = BLK([x_{cls}, x_1^k, \ldots, t_1^k, \ldots t_n^k]), \tag{1}$$

where $t_1^k \ldots t_n^k$ are $n$ prompts in the $k$th layer, $BLK$ represents the transformer block described above, and the $[\quad]$ represents the empty position which is left for the prompts in the next layer. In the last layer, the class token $x_{cls}$ is taken as the output and passes through a task-specific head combined with a *LayerNorm* and linear projector. In this paper, we also adopt the output of head from the classifier token $x_{cls}$ as the representation of our deep metric learning task.

## 3.2 PROXY-BASED DEEP METRIC LEARNING

Deep Metric Learning (DML) consider a set of data samples $X = \{x_i\}_{i=1}^N$ and their corresponding class labels $Y = \{y_i\}_{i=1}^N \in \{1, \ldots, C\}$; our goal is to learn a projection function $f_G : X \xrightarrow{f} \mathcal{X}$, which projects the input data samples to a hidden embedding space $\mathcal{X}$. The primal goal of DML is to refine the projection function $f_G(\cdot)$, which is usually constructed with CNN or Transformers as the backbone, to generate the projected features that can be easily measured with defined distance metric $d(x_i, x_j)$ based on the semantic similarity between image sample $I_i$ and $I_j$. We adopt the conventional distance metric $d(\cdot)$ as the *cosine similarity*. Before passing features to any loss, we use L2 normalization to eliminate the effect of differing magnitudes.

To further boost the learning efficiency, state-of-the-art methods set up a set of learnable representations $P = \{p_i \in \mathbb{R}^d\}_{i=1}^C$, named *proxies*, to represent subsets or categories of data samples. Typically, there is one proxy for each class, so the number of proxies is the same as the number of classes $C$. The proxies are also optimized with other network parameters. The first proxy-based method, Proxy-NCA (Movshovitz-Attias et al., 2017), or its improved version Proxy-NCA++ (Teh et al., 2020), utilizes the Neighborhood Component Analysis (NCA) (Goldberger et al., 2004) loss to conduct this optimization. The later loss Proxy-Anchor (PA) (Kim et al., 2020) alternatively sets the proxy as the anchor and measures all proxies for each minibatch of samples. We adopt the PA as our backbone loss since it shows robustness on various extended works (Venkataramanan et al., 2022; Roth et al., 2022; Zhang et al., 2022; Yao et al., 2022) Generally, for data sample set $X$ and proxy set $P$ the PA loss $\mathcal{L}_{proxy}(X, P)$ can be presented as,

$$\mathcal{L}_{proxy}(X, P) = \frac{1}{|P^+|} \sum_{p \in P^+} \log \left( 1 + \sum_{x \in X_p^+} e^{-\tau d(x,p)+\delta} \right) + \frac{1}{|P|} \sum_{p \in P} \log \left( 1 + \sum_{x \in X_p^-} e^{\tau d(x,p)+\delta} \right) \tag{2}$$

where $X_p^+$ denotes the set of positive samples for a proxy $p$; $X_p^-$ is its complement set; $\tau$ is the scale factor; and $\delta$ is the margin. Since the PA updates all approximations for each mini-batch, the model is more effective at learning the structure of samples beyond the mini-batch size.

# 4 PROPOSED METHODOLOGY

## 4.1 BASELINE ARCHITECTURE

In this paper, we propose optimizing the virtual prompts (VPT) to fine-tune the pre-trained ViT models for DML tasks. For our backbone model, we fix every parameter from the model except for the head projector. We send the prompts $t$ from the final block layer of the ViT to the head layer in dimension $D$, which is typically equivalent to the dimension of the hidden layer in the ViT. For DML tasks, we employ the Proxy-Anchor (PA) loss as our backbone loss, where the data samples are compared with the proxy anchor for each class as opposed to each other sample. Our framework and the baseline framework for DML tasks can be revealed in Figure 1.

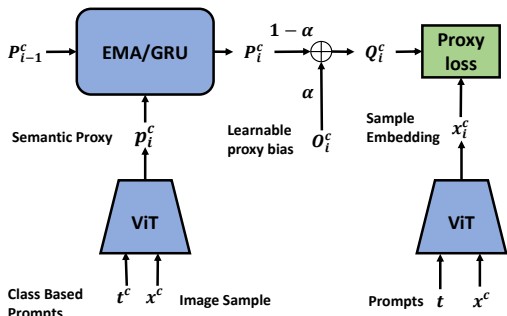

## 4.2 THE SEMANTIC PROXIES

One disadvantage of the proxy-based DML loss is that the proxies are randomly initialized and lack semantic information. Thus, the proxies are updated by comparing them to the sample embeddings that were not generated optimally during the early stages of training. To address this issue, we

Figure 2: Illustrate the architectures of our framework. Note that the parameters of the linear head are independent between the tower of the sample encoder and the semantic proxy. The tower of semantic proxy encodes all image samples of the same class $c$ and accumulates them into a single proxy for class $c$ with EMA or GRU.

attempt to initialize the proxies from the sample encoder with the same input data. However, this solution introduces a severe flaw: the proxies created by the data sample are close to or the same as the data samples from the same images. To deal with this problem, we included some extra learnable prompts dependent on sample class labels. In other words, for class $c$, we append a set of $m$ extra prompts $t^c \in \mathbb{R}^{m \times D}$ to the prompts of the data encoder in each layer. Figure 2 illustrates the details of our proposed architecture.

## 4.3 PROXY UPDATE AND INTEGRATION

Due to the variable quality of semantic proxies from each image in multiple batches at different training times, it is challenging to efficiently aggregate all of them into a single representation as the proxy for a typical class. One straightforward solution is inspired from the *exponential moving average* (EMA), in which we integrate the semantic proxies by adding each proxy to the representation of its associated class with a specific ratio. Assuming the $\mathcal{P}^c$ is the semantic proxy in class $c$, for each generated proxy $p_i^c$ in iteration $i$, we update $\mathcal{P}^c$ in the following manner.

$$\mathcal{P}_i^c = normal(\mathcal{P}_{i-1}^c + (1-\lambda)p_i^c) \tag{3}$$

where $\lambda$ is a hyperparameter between 0 and 1, and $normal$ represents the L2 normalization. We collect the newly arriving semantic proxy by updating with a larger ratio to depress the information from the previous iteration $(i-1)$ that needed to be better learned. Directly collecting the representations by EMA has several downsides. For example, the uniform decay to the vector may fail to capture the data structure within a short training time, and freshly produced proxies may contain noise irrelevant to the metric learning tasks.

To address these issues, we propose a novel and effective way to collect proxies in a recurrent manner, in which the semantic proxy is produced to be updated based on its current state and correlation to

the new proxy vector. Inspired by the *gated recurrent unit* (GRU) (Chung et al., 2014), we propose to update the $\mathcal{P}^c$ in iteration $i$ as follows:

$$z_i = \sigma(W_z p_i^c + U_z \mathcal{P}_{i-1}^c + b_z) \tag{4}$$

$$r_i = \sigma(W_r p_i^c + U_r \mathcal{P}_{i-1}^c + b_r) \tag{5}$$

$$\mathcal{N}_i^c = Relu(W_h p_i^c + r_i \odot U_h \mathcal{P}_{i-1}^c + b_h) \tag{6}$$

$$\mathcal{P}_i^c = normal((1 - z_i) \odot \mathcal{P}_{i-1}^c + z_i \odot \mathcal{N}_i^c) \tag{7}$$

where $W, U \in \mathbb{R}^{D \times D}$ are the learnable parameters to project the input $p_i^c$ and previous proxy $\mathcal{P}_{i-1}^c$ to a hidden space of dimension $D$, $b \in \mathbb{R}^D$ represents the bias vector, and $\odot$ denotes element-wise multiplication. In our GRU approach, $z_i \in \mathbb{R}^D$ in a range $(0, 1)$ represents the update gate at iteration $i$, and $r_i \in \mathbb{R}^D$ represents the reset gate at iteration $i$. The $\mathcal{N}_i^c$ is the candidate representation generated with reset gate $r_i$. Then the previous proxy $\mathcal{P}_{i-1}^c$ is further updated with the candidate $\mathcal{N}_i^c$ and gate $z_i$.

The GRU provides a more complex mechanism to update the accumulated proxy $\mathcal{P}_i^c$ as a hidden state, which is controlled with learnable parameters based on the new class proxy $p_i^c$ and the previous hidden state $\mathcal{P}_{i-1}^c$. The difference between our design and a normal GRU is that we propose applying the ReLU function rather than the Tanh function since we want to retain the updating of the vector scale. Note that we do not pass the gradient between the iterations or batches, and we feed the samples with the same class in each batch in random order to the accumulator.

After accumulating, we also add the original proxies, initialed randomly, as a bias with a certain ratio $\alpha$ to construct the output representation. For an output semantic proxy $\mathcal{P}_i^c$, we construct the output representation as follows:

$$\mathcal{Q}_i^c = normal((1 - \alpha)\mathcal{P}_i^c + \alpha \mathcal{O}_i^c) \tag{8}$$

where $\mathcal{O}_i^c$ represents the original proxies and $\mathcal{Q}_i^c$ represents the output representations for class $c$. $\alpha$ is pre-defined hyper-parameter. We further compare the performance of these two accumulation strategies in our experiments.

## 4.4 DISCUSSION

**Amount of Tunable Parameters** The amount of tunable parameters plays a crucial role in parameter-efficient research as it directly impacts the complexity of the training process. Based on the observation in the previous VPT work (Jia et al., 2022) where the prompts in the shallow layer have higher efficiency than the deep layer, we empirically set the number of prompts for each layer as a constant or decreasing number counting from the first layer. Specifically, from layer $i = 0$, we set the number of prompts for each layer to $n = N - \tau \times i$ where $\tau$ is the step that decreases the number of prompts based on the current layer. We also examine the impact of the number of tunable parameters in the Appendix.

**Memory Consumption** Memory usage is another important factor in evaluating parameter-efficient methods. In our approach, the number of prompts to construct semantic proxies increases linearly with the total number of classes in the downstream task. This will be a critical issue when the number of classes in this dataset grows. To reduce memory consumption, we develop an effective technique to organize prompts to conserve the GPU memory. In particular, we build a buffer in GPU memory to store the prompts that will be updated in its current batch. In other words, before running an iteration, we temporally shift the prompts of the semantic proxies in corresponding classes that are sampled in the batch from CPU memory to the GPU buffer and also update their statistic states to the training optimizer. Table 1 lists the comparison between ours and other methods in parameters and memory usage.

## 5 EXPERIMENTS
### 5.1 SETTINGS

In this paper, we focus on the standard vision transformer (ViT) of different scales (**ViT-S**, **ViT-B**) (Dosovitskiy et al., 2020). Without a specific claim, our backbone models in this section are pre-trained on ImageNet-21k (Ridnik et al., 2021). The selection of this pre-trained model was based

on its training on a sufficiently large dataset, which has been widely used in previous studies in the field of DML comparable to ours. Our method can be applied to larger pre-trained models, such as CLIP (Radford et al., 2021), which was trained on the gigantic Laion dataset. We also report the experiments with CLIP and other pre-trained models in our Appendix.

### 5.1.1 DATASETS AND METRICS

We utilize the conventional benchmarks CUB-200-2011 (**CUB200**) (Wah et al., 2011) with 11,788 bird images and 200 classes, and **CARS196** (Krause et al., 2013) that contains 16,185 car images and 196 classes. We also evaluate our method on larger Stanford Online Products (**SOP**) (Oh Song et al., 2016) benchmark that includes 120,053 images with 22,634 product classes, and In-shop Clothes Retrieval (**In-Shop**) (Liu et al., 2016) dataset with 25,882 images and 7982 classes. We follow the data split consistent with the standard settings of existing DML works (Teh et al., 2020; Kim et al., 2020; Ermolov et al., 2022). We also evaluate our methods on the **iNaturalist** (Van Horn et al., 2018), a larger dataset widely containing species in nature. We adopt the **Recall@K** proposed in existing works to evaluate ranking accuracy. In addition, we also evaluate it with Mean Average Precision at R (**MAP@R**), which is a more informative DML metric proposed by Musgrave et al. (2020). More details of our training progress and hyper-parameters searching are listed in the Appendix.

### 5.1.2 IMPLEMENTATION DETAILS

In this paper, we compare the existing parameter-efficient approaches and our baseline as follows:

1. **Fully Fine-tuning**: We fully update all parameters from the backbone, including the linear head with Proxy-Anchor loss.
2. **Linear Prob**: We only fine-tune the linear head with PA loss; all the rest of the parameters are frozen.
3. **Adapter**: We implement the Adapter by referring to the code released in the work (Chen et al., 2022a). We fine-tune the parameters belonging to the Adapter and the linear head while freezing all other parameters.
4. **Bitfit**: We fine-tuned all bias vectors and the linear head of the Transformer and fixed all other parameters, as proposed in Zaken et al. (2021).
5. **VPT**: This is the baseline setting following Jia et al. (2022) where we fine-tune the visual prompts, the linear head, and the learnable original proxies $\mathcal{O}^c$ while all other parameters are fixed.
6. **VPTSP-M** and **VPTSP-G**: The **VPT** with our proposed **S**emantic **P**roxies (VPTSP) is our framework, in which we generate semantic proxies by tuning class-based prompts and combining them with proxy bias. **-M** represents the setting we use EMA to accumulate the proxies, and **-G** represents the corresponding accumulation with GRU.

Following recent DML studies (Kim et al., 2020; Roth et al., 2022; Venkataramanan et al., 2022; Ermolov et al., 2022), we do some conventional data augmentation, which includes a random horizontal flip, cropping, and scaling the image to $224 \times 224$ pixels using bicubic interpolation. We also follow standard ViT configurations where the images are divided into $16 \times 16$ patches and keep the pre-trained [CLS] vector. We train our models in a machine that contains a single RTX3090 GPU with 24GB memory.

### 5.1.3 TRAINING DETAILS AND PARAMETER SEARCHING

We propose ViT-S/16 as our backbone for all the above settings to compare with other parameter-efficient methods. We fix the batch size to 64 with 2 samples in each class. We search the learning rate to optimize the Transformer encoder in a set of {0.05, 0.001, 0.0005, 0.0001, 0.00005}, and the learning rate to optimize the proxies in a set of {0.5, 0.1, 0.05, 0.01, 0.005}. For optimizers, we search within SGD (Robbins & Monro, 1951), Adam (Kingma & Ba, 2014), and AdamW (Loshchilov & Hutter, 2017). Specifically, we search the architecture for both the Adapter and VPT, including the mid-dimension, position, and number of applied layers for the Adapter and the number of prompts and applied layers for VPT and our framework. More details of our search range and selected parameters can be referred to in the Appendix.

| Method | Compare Architecture and Performance on CUB200 | | | |
| --- | --- | --- | --- | --- |
| | Parameter(tunable) | Memory | R1 | MAP@R |
| Full Fine-tuning (PA) | 30.2M (100%) | 6.0G (100%) | 85.5 | 51.1 |
| Linear Prob | 0.19M (0.63%) | 1.7G (28.3%) | 84.9 (-0.6) | 48.1 (-3.0) |
| BitFit | 0.26M (0.86%) | 4.4G (73.3%) | 85.7 (+0.2) | 51.3 (+0.2) |
| Adapter (L:7,d:256) | 1.57M (5.2%) | 1.9G (31.7%) | 83.1 (-2.4) | 46.4 (-4.7) |
| Adapter (B) (L:1,d:256) | 0.46M (1.52%) | 2.4G (40%) | 85.8 (+0.3) | 49.7 (-1.4) |
| VPT (L:12,N:10) | 0.23M (0.76%) | 2.2G (36.6%) | 85.1 (-0.4) | 51.2 (+0.1) |
| VPT (B) (L:12,N:10) | 0.30M (0.99%) | 2.2G (36.6%) | 85.7 (+0.2) | 51.6 (+0.5) |
| VPTSP-M (ours) | 0.60M (1.9%) | 2.5G (41.7%) | 85.4 (-0.1) | 51.4 (+0.3) |
| VPTSP-M (B) (ours) | 0.67M (4.5%) | 2.6G (43.3%) | 85.9 (+0.4) | 51.8 (+0.7) |
| VPTSP-G (ours) | 1.48M (4.9%) | 2.5G (41.7%) | 86.1 (+0.6) | 52.1 (+1.0) |
| VPTSP-G (B) (ours) | 1.56M (5.2%) | 2.6G (43.3%) | **86.6 (+1.1)** | **52.7 (+1.6)** |

Table 1: Compare our framework and other parameter-efficient methods on the CUB200 dataset. For each approach, we look for hyperparameters in the range mentioned above. The **Full Fine-tuning (PA)** approach serves as our baseline, which builds upon the PA loss (Kim et al., 2020). **L:** denotes the number of layers to which the Adapter or VPT should be applied, **d:** represents the Adapter's mid-dimension, and **N:** represents the number of prompts applied in each layer. Furthermore, **-M** represents our approaches that used EMA fusion, **-G** represents our methods that used GRU fusion, and **(B)** represents applying the BitFit method. We introduce memory checkpoints in the transformer block layer for both Adapter and VPT to save memory. We apply the memory buffer indicated above for our proposed method.

## 5.2 QUALITATIVE RESULTS

**Comparing with Other Parameter-efficient Methods.** The brief comparison results can be revealed in Table 1. It has been discovered that BitFit has a universal advantage for all methods, resulting in a $2.7pp$ gain in R@1 and a 3.3pp increase in MAP@R for the Adapter. Additionally, it leads to a $0.6pp$ improvement in R@1 and a $0.4pp$ increase for VPT. We found that the linear probing has a slightly weak performance compared with full fine-tuning ($-0.6pp$ on R@1). Compared to Adapter and VPT, we find that VPT benefits more for DML tasks and outperforms the full fine-tuning method (when combined with BitFit). This is the primary reason we chose VPT as the foundation for designing the framework for parameter-efficient DML. Compared with our method, we demonstrate that our semantic proxies significantly improve compared with the baseline VPT method with $1.5pp$ on R@1 and $0.8pp$ on MAP@R. Our method has $1.7pp$ better performance on the R@1 metric than linear probing with slightly more parameters. Our approach also has better performance than the Adapter approaches.

| Method | Settings | CUB-200 | | | CARS-196 | | | SOP | | | InShop | | |
| --- | --- | --- | --- | --- | --- | --- | --- | --- | --- | --- | --- | --- | --- |
| Name (Batch Size) | Arch/Dim | R@1 | R@2 | R@4 | R@1 | R@2 | R@4 | R@1 | R@10 | R@100 | R@1 | R@10 | R@20 |
| Hyp-ViT (882/900) | ViT-S16/384 | 85.6 | 91.4 | 94.8 | 86.5 | 92.1 | 95.3 | **85.9** | **94.9** | **98.1** | 92.5 | **98.3** | **98.8** |
| VPT-Base (64/32) | ViT-S16/384 | 85.1 | 91.1 | 94.0 | 86.3 | 91.8 | 95.5 | 82.1 | 92.5 | 97.1 | 88.4 | 97.5 | 98.4 |
| VPTSP-M (64/32) | ViT-S16/384 | 85.4 | 91.2 | 94.6 | 86.8 | 92.0 | 95.5 | 82.8 | 93.1 | 97.3 | 90.8 | 97.7 | 98.6 |
| VPTSP-G (64/32) | ViT-S16/384 | **86.6** | **91.7** | **94.8** | **87.7** | **93.3** | **96.1** | 84.4 | 93.6 | 97.3 | 91.2 | 97.6 | 98.4 |
| RS@K (>200) | ViT-B16/512 | – | – | – | 89.5 | 94.2 | 96.6 | **88.0** | **96.1** | **98.6** | – | – | – |
| VPTSP-G (64/32) | ViT-B16/512 | **88.5** | **92.8** | **95.1** | **91.2** | **95.1** | **97.3** | 86.8 | 95.0 | 98.0 | **92.5** | **98.2** | **98.9** |

Table 2: Comparison with the state-of-the-art full fine-tuning methods, including Hyp-ViT (Ermolov et al., 2022) and RS@K (Patel et al., 2022) applying ViT on the conventional benchmarks pre-trained on ImageNet-21K. The second column shows the architecture of the backbone and the feature dimension we selected to compare. We also indicate the batch size for training in the brackets. ViT-S16 and ViT-B16 represent the *small* and *basic* size of ViT with patch size $16 \times 16$. We shade our proposed method, and the best under the same pretraining and architecture are bold.

**Comparing with state-of-the-art.** We further compare the performance of our method with the state-of-the-art methods based on the ViT that was pre-trained on ImageNet-21K as listed in Table 2 and 3. For both the CARS196 and the CUB200 datasets, our method reaches better R@1, which has significantly improved over the previous state-of-the-art on the same ViT-S pre-trained model. Also, our results with ViT-B/16 outperform the RS@K (Patel et al., 2022), the previous state-of-the-art work on ViT-B. For CUB200, our framework outperforms the previous state-of-the-art Hyp-ViT

| Conventional Evaluation on iNaturallist | | | | | | |
|---|---|---|---|---|---|---|
| Method | Architecture | R@1 | R@4 | R@16 | R@32 | MAP@R |
| SAP (Brown et al., 2020) | ViT-B16/512 | 79.1 | 89.0 | 94.2 | 95.8 | – |
| Recall@K (Patel et al., 2022) | ViT-B16/512 | 83.9 | 92.1 | 95.9 | 97.2 | – |
| VPTSP-G (ours) | ViT-B16/512 | **84.5** | **92.5** | **96.4** | **97.5** | **48.1** |

Table 3: Experimental results on *iNaturalist*. We compare with the state-of-the-art method (Patel et al., 2022) on this dataset under the same pre-trained dataset and architecture. We bold the higher results in the same setting and metric.

(Ermolov et al., 2022). For larger datasets, SOP, InShop, and iNaturalist, we also have a comparable or better result with the existing state-of-the-art full fine-tuning works. Notably, specific datasets, such as SOP, are marginally enhanced by our proposed framework. This phenomenon can be attributed to the greater number of classes, which complicates the process of quickly optimizing the parameters of each proxy. Note that most existing DML methods substantially rely on the full fine-tuning of a pre-trained model with different sampling or optimization strategies. As a result, their computing costs and runtime latencies are close to our full fine-tuning baseline, which is much larger than our proposed method. Compared to existing DML approaches, our proposed method demonstrates reductions in computational cost, batch size, and memory usage while achieving comparable DML performance.

| | CUB200 | | CARS196 | |
|---|---|---|---|---|
| | R@1 | MAP@R | R@1 | MAP@R |
| Full Fine-tuning (PA) | 85.5 | 51.1 | 86.7 | 28.7 |
| VPT+Bias (**Baseline**) | 85.7 | 51.6 | 86.3 | 27.6 |
| VPTSP (EMA) | 85.4 (-0.3) | 51.4 (-0.2) | 86.4 (+0.1) | 28.3 (+0.7) |
| VPTSP (EMA) +Bias | 85.9 (+0.2) | 51.8 (+0.2) | 86.8 (+0.5) | 28.7 (+1.1) |
| VPTSP (GRU) | 86.1 (+0.4) | 52.1 (+0.5) | 87.1 (+0.8) | 28.6 (+1.0) |
| VPTSP (GRU) + Bias | **86.6 (+0.9)** | **52.7 (+1.1)** | **87.7 (+1.4)** | **28.9 (+1.3)** |

Table 4: Ablation of components of our framework on CUB200 and CARS196. **VPT** is the conventional virtual prompts approach, and **VPTSP** is our proposed VPT with the Semantic Proxies method. **Bias** represents the Bitfit for all approaches, and all settings we show here are already combined with linear probing.

## 5.3 ABLATION STUDIES

We analyze the contribution of each component of our proposed framework based on the results of CUB200 and CARS198. As listed in Table 4, our GRU accumulation improves the performance on both R@1 and MAP@R, which conducts the efficiency of the recurrent accumulation in the training progress. Also, it is clear that our proposed semantic proxies significantly improve the performance of both datasets. We also analyze the impact of the hyperparameters, including the fusion ratio $\lambda$ for EMA, the ratio $\alpha$ for fusion with the proxy bias, the number of the prompts, and the layers the prompts should append to in our Appendix.

## 6 CONCLUSION

In this paper, we investigate the application of parameter-efficient optimization strategies to various deep metric learning (DML) tasks. The performance of existing parameter-efficient strategies on DML benchmarks is first evaluated. Then, we enhance the efficacy of metric learning using the most effective technique, visual prompt tuning (VPT). Specifically, we develop a quick and efficient framework based on the conventionally successful proxy-based DML paradigm, in which we improve the representation capability and metric learning performance by integrating the semantic information from the pre-trained ViT that has been modified with class-based prompts. Our evaluation demonstrates the learning effectiveness of our method, and our extensive experiments reveal a superior intent to enhance metric learning and retrieval performance further while maintaining a small number of tuning parameters.

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

APPENDIX: LEARNING SEMANTIC PROXIES FROM VISUAL PROMPTS FOR PARAMETER-EFFICIENT TUNING IN DEEP METRIC LEARNING

## A EXTENSIVE DISCUSSIONS

In this section, we explain and discuss the insight behind our design of semantic proxies for the parameter-efficient DML.

### A.1 WHY DO THE SEMANTIC PROXIES BENEFIT THE PROXY-BASED DML?

Proxy-based DML techniques have gained widespread acceptance because of their greater performance when compared to other current DML losses (Kim et al., 2020; Roth et al., 2022; Venkataramanan et al., 2022; Zhang et al., 2022; Yang et al., 2022). However, its learning process continues to be hindered by a basic issue: the huge distribution gap between the initialized proxies and the data samples, in which the proxies are initialized randomly with no prior knowledge of their distribution (Ren et al., 2024). This would result in low convergence efficiency and bias in the learning process. To address this issue, we offer semantic proxies, in which semantic information is embedded into the proxies from the beginning. Even if the class-based prompts are not fully trained at the start, the pre-trained ViT still encodes rich semantic information to the initialized proxies, improving the efficiency of learning data embeddings. For example, the visualized comparison between the semantic and original proxies can be referred to in Figure. 4.

### A.2 WHY DO WE EXPLORE THE FINE-TUNING METHOD SPECIFICALLY FOR DML TASKS?

Given that existing works have explored several fine-tuning methods for CV tasks. However, the evaluation of their performance for deep metric learning and image retrieval tasks still needs to be completed. Our proposed method achieves a higher computational efficiency than full fine-tuning methods and improves the overall accuracy. As highlighted in our paper, our technique has achieved state-of-the-art results in several benchmarks. In times dominated by large models, our contribution is invaluable to model users who might need more computational resources but hope to achieve the best performance in their local image retrieval tasks. We remain convinced of the relevance and value of investigating faster and more effective alternative fine-tuning methods, especially for specific local image retrieval tasks. While the design of prompts for tuning has been a subject of recent research, there remains a gap in the exploration of prompt tuning methods tailored for deep metric learning and image retrieval. Our work draws inspiration from prompt tuning methodologies, but our primary emphasis lies in elevating image retrieval performance. We achieve this by innovating semantic proxies that mitigate the shortcomings of existing full fine-tuning-based deep metric learning techniques.

### A.3 WHY DO WE COMBINE THE ORIGINAL PROXIES TO THE SEMANTIC PROXIES AS BIAS?

One limitation of our suggested semantic proxies is the restricted range of parameters that can be updated for each class in batch. We would only update the parameters of the semantic proxies whose classes are sampled into the batch in each iteration, but the proxy-based loss compares each sample with proxies from all classes. As a result, the classes that were not sampled will need to be updated or have *semantic meaning shift* compared to the classes in the batch. When the number of classes is vast, updating all semantic proxies in each cycle might be difficult. To address this issue, we propose combining the original proxies, which can readily be updated outside of the mini-batch, as a bias when the semantic proxies cannot be updated in time. In other words, inside the mini-batch, we would update **both the semantic proxies and the bias**, whereas, outside the mini-batch, we would **update the bias**. This would balance the trade-off between each batch's parameter update frequency and training delay.

### A.4 WHAT IF WE DIRECTLY ACCUMULATE THE SAMPLE REPRESENTATIONS AS SEMANTIC PROXIES?

This is one of our early, primitive attempts. It is valid when the sample representations are directly accumulated as semantic proxies (without fine-tuning using virtual prompts). However, we discovered

| Settings | CUB200 | | CARS196 | |
|---|---|---|---|---|
| | R@1 | MAP@R | R@1 | MAP@R |
| Sample | 85.7 | 51.2 | 86.2 | 28.9 |
| Encoder | 86.1 | 51.5 | 86.9 | 28.4 |
| Fixed Encoder | 85.4 | 50.5 | 86.4 | 28.7 |
| VTPSP-G (ours) | **86.6** | **52.7** | **87.7** | **28.9** |

Table 5: Experimental results on the settings to evaluate the efficiency of our proposed class-based virtual prompts. Note that we do not consider the contribution of proxy bias in this comparison.

| Method | Arch | Parameters | R@1 | R@2 | R@4 | MAP@R |
|---|---|---|---|---|---|---|
| ProxyNCA++ (Teh et al., 2020) | R50 | 26.5M | 69.0 | 79.8 | 87.3 | – |
| NIR (Roth et al., 2022) | R50 | 26.5M | 69.1 | 79.6 | – | – |
| XBM+PA (Wang et al., 2020) | BN | 12M | 65.8 | 75.9 | 84.0 | – |
| PA (vanilla) (Kim et al., 2020) | R50 | 26.5M | 69.7 | 80.0 | 87.0 | 26.5 |
| Hyp (Ermolov et al., 2022) | ViT-S/16 | 30.2M | 85.6 | 91.4 | **94.8** | – |
| Full fine-tuning (PA) | ViT-S/16 | 30.2M | 85.1 | 91.1 | 94.0 | 51.1 |
| VPTSP (ours) | ViT-S/16 | **1.56M** | **86.6** | **91.7** | **94.8** | **52.7** |

Table 6: In this study, we present a detailed comparison with state-of-the-art works in the field, as outlined in the CUB200. Specifically, the second column lists various backbone architectures: **R50** denotes ResNet50, **BN** refers to InceptionBN, and **ViT-S/16** signifies the small Vision Transformer utilizing $16 \times 16$ patches. However, it is essential to note that this comparison may not be entirely equitable due to differences in the backbone architectures and the datasets used for pretraining.

that this method cannot resolve the change in distribution between the pre-trained and fine-tuning datasets. The semantic proxies built in this situation can benefit the sample encoder during the early training stage but will mislead the encoder training later. To further demonstrate the contribution of our class-based prompts, we propose an extended experiment with the following settings:

1. **Sample**: Semantic Proxies accumulated from encoder samples (no class-based prompts).

2. **Encoder**: Semantic Proxies from encoder samples with same prompts from encoder.

3. **Fixed Encoder**: Proxies from encoder samples with same prompts from encoder, but do not pass gradient.

The Setting **Sample** is proposed to evaluate the semantic shift we discussed. In Setting **Encoder**, we improve the weakness of setting **Sample** by involving a trainable sample encoder. For Setting **Fixed Encoder**, we evaluate whether the learning signal from the proxy side limits the capability of representations. The result of this comparison can be referred to Table 5. The results demonstrate that generating the semantic proxies with the original ViT encoder does not have enough capability. Generating proxy from the same trainable encoder has higher R@1 but has low MAP@R, which reflects overall retrieval performance. Compared with the fixed encoder, we also conclude that the gradient from the samples and proxies encoder does not have the same direction. Based on this conclusion, our proposed framework, which provides an isolated proxy encoder for each class, is the optimized solution compared with these settings with simply a few more parameters.

## A.5 COMBINE THE VPT AND ADAPTER

Conducting the settings where the VPT and Adapter are combined is straightforward. Unfortunately, we did not find any benefit in combining these two architectures. Only with minimal parameters, Adapter has close performance with the pure VPT approach. This may be because the Adapter is unsuitable for DML tasks where the extra trainable parameters need to provide more learning capability while increasing the training difficulties.

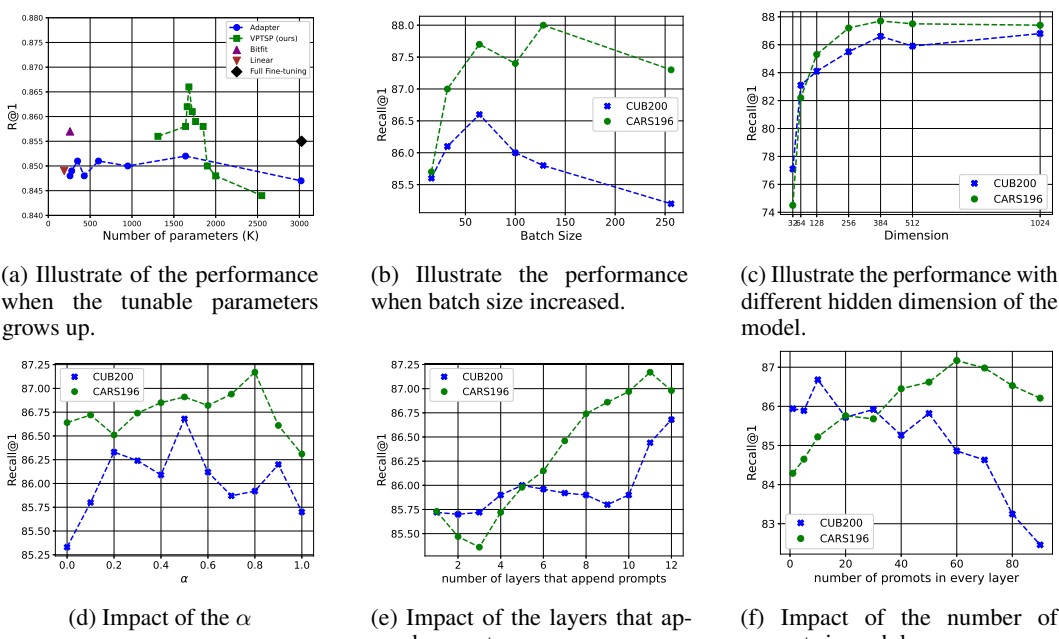

(a) Illustrate of the performance when the tunable parameters grows up.

(b) Illustrate the performance when batch size increased.

(c) Illustrate the performance with different hidden dimension of the model.

(d) Impact of the $\alpha$

(e) Impact of the layers that append prompts

(f) Impact of the number of prompts in each layer

Figure 3: We compare the performance with different tunable parameters in Figure (3a). Our VPTSP has an optimal range in which the amount and layers of prompts are appropriate for fine-tuning the model. When the number of prompts increases, the learnable capacity is reached, and the extra prompts will not be tuned well. Figure (3b) shows that our proxy-based DML technique has an appropriate batch size range. When the batch size increases too large, the overall performance suffers. In Figure (3c), we also compare the various dimension values with similar features to the batch size. We also illustrate the impact of $\alpha$, which is the fusion ratio between the original proxies and our semantic proxies, the number of layers that we apply the prompts as additional, and the number of prompts in each layer in Figure (3d) (3e) and (3f)

## A.6 IMPACT OF HYPEPARAMETERS

We also analyze the impact of the hyperparameters, including the fusion ratio $\lambda$ for EMA, the ratio $\alpha$ for fusion with the proxy bias, the number of prompts, and the layers where the prompts are set up. As illustrated in 3, when the number of prompts is larger than a limit, which varies between datasets, the overall performance starts to drop. The layers with prompts also have similar properties. These observations are consistent with other vision tasks reported in previous works (Jia et al., 2022).

## B EXTENTED EXPERIMENTS

### B.1 COMPARE THE METHOD TO INTEGRATE PROXIES

In this section, we compare the performance of different RNN architectures and other networks to integrate the semantic proxies. We searched the architecture of recurrent units in a small space and compared the recurrent neural network with cross-attention networks. For the cross-attention approach, we only integrate representations within the mini-batch. A critical weakness of attention and cross-attention in our case, where we integrate the representations from different batches and iterations, is that they have to integrate all data within the same iteration. If we add too much data in the same iteration for cross-attention, there will be external time complexity and memory. Table 7 lists the evaluation results on CUB200, and we can see that our proposed GRU approach performs better.

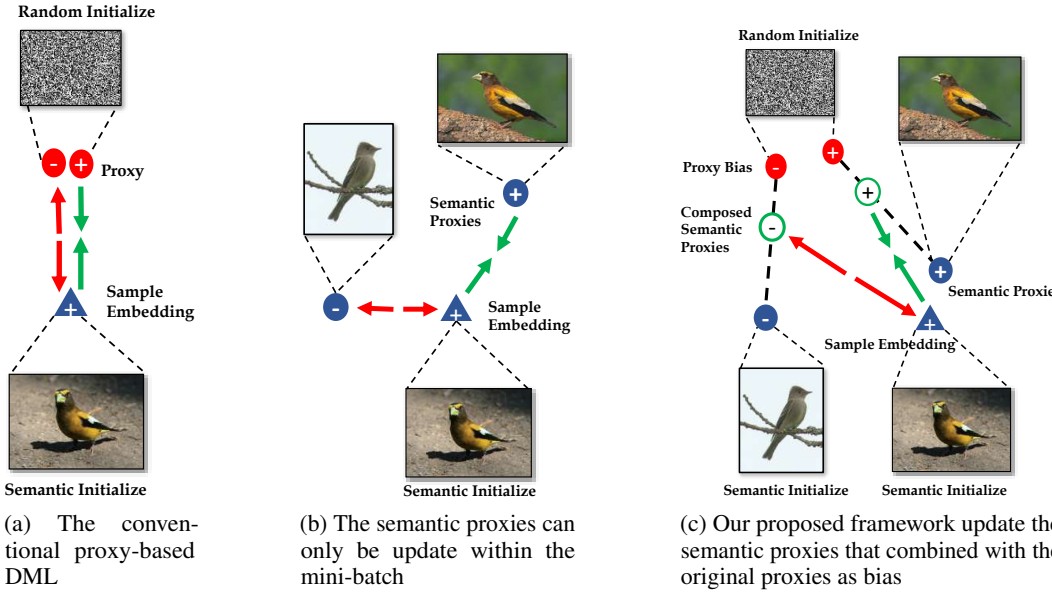

(a) The conventional proxy-based DML

(b) The semantic proxies can only be update within the mini-batch

(c) Our proposed framework update the semantic proxies that combined with the original proxies as bias

Figure 4: Illustrate the space of the original proxy-based DML, our semantic proxy, and our proposed metric space. The triangles represent the generated image embeddings, and the circles represent the proxy. The + represents the positive (in the same class), and the − represents the negative (not in the same class). The green circle represents our composed semantic proxies.

| Integration Architecture | R@1 | MAP@R |
|---|---|---|
| LSTM | 85.1 | 49.8 |
| RNN Unit | 84.6 | 49.2 |
| Cross Attention | 82.2 | 47.8 |
| GRU with Tanh | 85.7 | 51.4 |
| GRU with Relu | 86.1 | 52.1 |

Table 7: Comparing the network architecture to integrate semantic proxies on CUB200.

## B.2 EXTENSION ON PRE-TRAINING MODELS

**ImageNet-1K** pretraining is widely applied to fine-tun in CNN-based architecture. Although people started to focus on larger datasets for pre-training after moving to Transformers, there are still a few high-quality pretraining strategies and models in ImageNet-1K that show promising results in CV tasks. Here, we also provide a comparison of DML performance between parameter-efficient fine-tun methods on two popular pre-trained models, DeiT (Touvron et al., 2021) and DINO (Caron et al., 2021), as an additional reference.

**MAE** (He et al., 2022) is an interesting unsupervised learning method where the representations are directed by guessing the blank patches. Since it is also pre-trained on ImageNet-1K, we expect it to perform better than other ImageNet-1K pretraining methods.

**CLIP Vision Tower** (Radford et al., 2021) is a well-known large-scale model pre-trained on a massive dataset containing billions of image-text pairs. We test our method on the single vision tower of CLIP to evaluate the performance of our method in a large-scale model. The evaluation result can be referred to in Table 8.

The results are presented in Table 8. We believe that the VPT approach and other parameter efficiency methods are influenced by the differences in distributions between the pre-trained and fine-tuning datasets. In contrast to the large datasets, smaller pretraining datasets, such as ImageNet-1K, often show larger distributional variance. Consequently, the full fine-tuning strategies have better performance on smaller pretraining datasets. However, the overall performance of the full fine-tuning methods on small pretraining datasets is still limited compared with larger pretraining datasets of their limited general ability.

| Architecture | Model | Pre-training Set | CUB200 | CARS196 | SOP | InShop |
|---|---|---|---|---|---|---|
| Vit-S/16 | DINO | ImageNet 1K | 82.1 | 86.9 | 82.6 | 90.1 |
| Vit-S/16 | DeiT | ImageNet 1K | 80.2 | 84.5 | 81.2 | 88.7 |
| Vit-B/16 | MAE | ImageNet 1K | 85.1 | 83.2 | 77.3 | 90.3 |
| Vit-L/14 | CLIP (vision) | Laion2b | 86.7 | 97.4 | 90.1 | 96.5 |

Table 8: We list the supplemental results from our framework with different architecture and pre-trained models. We report the R@1 result for each pre-trained model and dataset.

| Search Attribute | Search Range |
|---|---|
| Global Parameters | |
| optimizer | {SGD, Adam, AdamW} |
| Batch Size | {32, 64} |
| Number of Sample per Class (n) | {1, 2} |
| $lr$ | {0.05, 0.01, 0.005, 0.001, 0.0005, 0.0001} |
| $lr$ for proxy bias | {0.5, 0.1, 0.05, 0.01, 0.005, 0.001} |
| Adapter | |
| Number of Layers (L) | {1,2, ..., 11, 12 } |
| mid-dimension (d) | {1, 4, ..., 512} |
| Adapter Position | Sequential / Parallel |
| Initialization | uniform / kaiming uniform |
| Adapter Position | Pre / Post |
| VPT | |
| Number of Layers (L) | {1,2, ... 11, 12 } |
| Number of Prompts (N) | {1, 5, 10, 30, 50, 70, 90} |
| Decreasing Step ($\tau$) | {0, 2, 5, 10} |
| Our Method | |
| Number of Layers for Class Based Prompts (CLS-L) | {2, 4, 6, 8, 10, 12} |
| Number of Class Based Prompts (CLS-N) | {1, 3, 5, 10, 40, 80} |
| Ratio between Semantic Proxy and Bias ($\lambda$) | {0.1,0.2,...,0.9} |

Table 9: We list the search space for each method we compare with our framework. For each method, we first search a pair of proper $lr$ and $lr$ for proxy and then the corresponding architectures with a grid based on the $lr$ and $lr$ for proxy.

## C  ANALYSIS ON LATENCY

To compare the training efficiency between parameter-efficient methods in DML tasks more intuitively, we compare the overall running time of each iteration under the same machine. For all running times, we average the running time over 100 trials. From the results in Table 10, we can conclude that our proposed framework takes slightly more time than other parameters, but we still have a faster training speed than the full fine-tuning process. It is a fair trade from the training latency (training complicity) to the overall performance.

| Methods | Latency (B=32) | Latency (B=64) |
|---|---|---|
| Full Fine-tuning (PA) | 191ms | 345ms |
| linear | 115ms | 120ms |
| BitFit | 130ms | 210ms |
| Adapter | 120ms | 190ms |
| VPT | 140ms | 226ms |
| VPTSP | 166ms | 260ms |

Table 10: List of the running time of a single forward-backward step on Cars196 with different batch sizes (average over 100 times). The **Full Fine-tuning (PA)** approach serves as our baseline, which builds upon the Proxy-Anchor method (Kim et al., 2020). However, it incorporates a ViT backbone that received pre-training on the larger ImageNet-21k dataset.

## D    LIMITATION AND BOARDER IMPACT

**Limitation** Our baseline and proposed framework clearly do not outperform the most recent state-of-the-art on SOP and InShop. This is because there are many more classes in these two more extensive datasets. It will be challenging to properly train the parameters based on each class due to the large number of classes. This is a widespread flaw with various proxy-based methods, to be sure. By attempting to reduce the parameters for proxy and semantic proxies or finding other potential ways to boost their learning efficiency, we will strive to address this challenge in our subsequent work. Because the state-of-the-art full fine-tuning works (Ermolov et al., 2022) propose to train their model with distributed settings on multiple GPUs with large memory (e.g., $8 \times$A100) for a long time, we still propose a framework with close results but with much smaller batch size, faster learning efficiency, and lower GPU cost which can be efficiently completed on a single GPU.

**Boarder Impact** We thoroughly compare the most popular parameter-efficient algorithms on the essential deep metric learning tasks as the first effort to investigate the parameter-efficient framework for retrieval tasks. We also suggest a new framework based on the VPT to address the issue with existing proxy-based approaches where the initialed proxies lack semantic information. We expect that these contributions will give ideas and inspiration for future efforts that will investigate how to efficiently adapt large-scale models to deep metric learning or image retrieval applications.

