# OpenReview forum: "Learning Semantic Proxies from Visual Prompts for Parameter-Efficient Fine-Tuning in Deep Metric Learning"
_ICLR.cc/2024/Conference — ICLR 2024 poster_

### Official Review · Reviewer_1Fvy · 2023-10-30

**Soundness:** 3 good
**Presentation:** 3 good
**Contribution:** 3 good
**Rating:** 6
**Confidence:** 3

**Summary:**

This paper addresses the challenge of adapting pre-trained models for Deep Metric Learning (DML) tasks while preserving prior knowledge. Existing solutions often fine-tune models on standard image datasets, making it difficult to apply them to local data domains. The paper introduces a novel approach, the Visual Prompts (VPT) framework, which augments the conventional proxy-based DML method. VPT optimizes visual prompts for each class by integrating semantic information from input images and Vision Transformers (ViT). The experimental results demonstrate that this approach outperforms existing methods in DML benchmarks, achieving comparable or better performance with minimal parameter fine-tuning. This parameter-efficient technique offers a promising avenue for improving DML without requiring extensive model retraining.

**Strengths:**

- The paper introduces a novel approach to Deep Metric Learning (DML) by proposing the Visual Prompts (VPT) framework. This framework addresses the challenge of fine-tuning pre-trained models for DML while incorporating semantic information and optimizing visual prompts. This approach represents an innovative combination of existing ideas, extending the traditional proxy-based DML paradigm with an efficient and effective method.

- The paper is well-written and clearly articulates the motivation, methodology, and experimental results. The authors effectively communicate their approach, making it accessible to a broad audience of readers, including those familiar with DML and those new to the field. The paper's clarity enhances its potential for adoption and understanding by the research community.

- The proposed framework has the potential to streamline and improve DML tasks, as it achieves comparable or better performance compared to full fine-tuning approaches, all while adjusting only a small percentage of total parameters.

**Weaknesses:**

- The paper lacks a comprehensive comparison with existing DML methods, especially those using transformer based backbone. Additionally, it would be beneficial to compare the proposed Visual Prompts (VPT) framework not only against full fine-tuning but also against other state-of-the-art parameter-efficient methods. This would provide a more complete assessment of its effectiveness and novelty.

**Questions:**

See weakness.

---

> ### Author Response · Authors · 2023-11-22
> **We thank you for your review**
>
> We are deeply grateful for the time and effort you invested in reviewing our manuscript. Your insightful comments and the connection you drew between our methods and existing DML ideas are highly appreciated.
>
> **Addressing the Need for Comprehensive Comparison with Existing DML Methods**
>
> In alignment with our response to **Reviewer 1 (hvFX)**, we wish to further clarify the baseline methodologies used in our study. The "full fine-tuning" baseline, as discussed in both our main paper and Appendix, is based on the proxy-anchor approach [1], which is recognized as one of the state-of-the-art methods in DML. Additionally, we have conducted comparisons with other leading approaches, such as Hyp-ViT [2] and RS@k [3], which are also based on ViT architectures and considered state-of-the-art.
>
> It is important to note that all these state-of-the-art methods [1][2][3], while varying in their optimization loss, share a commonality in their computational costs, as they are all fundamentally full fine-tuning methods. To provide a more detailed comparison, we have included an additional analysis in Table 6 in the Appendix. This table compares our methods with those mentioned and other existing DML methods, offering a comprehensive comparative view.
>
> For an in-depth understanding of these comparisons and the rationale behind our approach, we kindly refer you to our detailed response to **Reviewer 1 (hvFX)**, where we elaborate on these points further.
>
>
> **Addressing the comparison with other parameter-efficienct fine-tuning methods**
>
> We would like to draw your attention to our comprehensive analysis comparing our VPT method with several leading parameter-efficient techniques in the DML benchmark. This detailed comparison is outlined in Section 5.1.2 of our paper. The results of this comparative study are showcased in Table 1 of the main manuscript, where we assess and compare the performance of each method in terms of DML metrics (specifically R@1 and MAP@R), the number of adjustable parameters, and memory usage.
>
> To ensure a fair and unbiased comparison, as elaborated in Section 5.1.3, we conducted an extensive hyperparameter search for each method. This rigorous approach ensures that our comparative analysis is not only comprehensive but also equitable, taking into account the varied characteristics and capabilities of each parameter-efficient method under consideration.
>
> We trust that these clarifications address your concerns regarding the comparative analysis of our methods with other DML approaches. Your constructive feedback is invaluable in our pursuit of continual improvement of our manuscript.
>
>
> [1] Kim, Sungyeon, et al. "Proxy anchor loss for deep metric learning." CVPR 2020.
>
> [2] Ermolov, Aleksandr, et al. "Hyperbolic vision transformers: Combining improvements in metric learning." CVPR 2022.
>
> [3] Patel, Yash, Giorgos Tolias, and Jiří Matas. "Recall@ k surrogate loss with large batches and similarity mixup." CVPR. 2022.

---

### Official Review · Reviewer_HvDC · 2023-11-01

**Soundness:** 4 excellent
**Presentation:** 3 good
**Contribution:** 4 excellent
**Rating:** 8
**Confidence:** 4

**Summary:**

This paper proposes an efficient method for fine-tuning the pre-trained models for the DML image retrieval tasks. The authors propose a method based on visual prompts (VPT) to partially fine-tune the model instead of tuning all parameters. Based on the proxy-based DML methods, they also initial the proxies with input images and visual prompts based on classes.

**Strengths:**

1 The contribution is solid. As far as I know, this may be the first work to explore the parameter-efficient (PEFT) methods specifically for deep metric learning.

2 Novelty is fine. Although the VPT fine-tuning is not something new, the local design to improve it specifically for DML is interesting and novel.

3 The writing is clear and easy to follow.

4 Experiments is well organized and convincible.

The authors evaluate their method widely on popular DML datasets, and the results seem to be strong and solid. They also comprehensively compare other PEFT methods on DML datasets, which might be interesting to the community.

**Weaknesses:**

I didn't see significant weakness. Based on the limited results on some datasets compared with SOTA, I hope the authors can provide more analysis and possible ideas to improve it.

It seems the results are good on large pre-training datasets but fair on small pre-training. Could you explain more about this phenomenon?

It is suggested to move some results in the Appendix, like results on other datasets and pre-trained models, to the main paper.

Minor:
Some typo or grammar errors:
"which includes a random horizontal flip, cropping at random..."
"we found that it linear probing..."

Some recent works missing:

Wang et al. "Deep Factorized Metric Learning." CVPR 2023
Kim et al. "HIER: Metric Learning Beyond Class Labels via Hierarchical Regularization" CVPR 2023
Kotovenko et al. "Cross-Image-Attention for Conditional Embeddings in Deep Metric Learning", CVPR 2023

**Questions:**

Please see the weaknesses.

---

> ### Author Response · Authors · 2023-11-22
> **Expression of Gratitude for Your Detailed Review**
>
> We are immensely grateful for the time and effort you have devoted to reviewing our manuscript. Your acknowledgment of our paper's contribution and novelty, coupled with your valuable insights and suggestions, is highly appreciated.
>
> **Addressing Limited Results on Some Datasets Compared with SOTA**
>
> In Section 5 of the Appendix, we have openly discussed the limitations of our proposed VPT methods. The suboptimal performance on the SOP and InShop datasets, which feature a higher number of classes and proxies, is a focus of our ongoing research and improvement. This admission underlines our commitment to transparency and the continuous development of our work.
>
> **Performance on Large vs. Small Pre-training Datasets**
>
> As elaborated in the Experimental section, we posit that the performance of VPT and other parameter efficiency methods is influenced by the distributional variances between pre-training and fine-tuning datasets. Smaller pre-training datasets like ImageNet-1K often display a more significant variance in distribution compared to target datasets. This observation is consistent with prior findings in VPT research [1], lending support to our analysis.
>
>
> **Reorganization of Results and Appendix Content**
>
> In response to your suggestion, we have included the results from the iNaturalist dataset in the main paper (now in Table 3). Due to space and format constraints, we are unable to relocate more results from the Appendix to the main text. We believe that the current structure of the manuscript strikes a balance between detail and conciseness, effectively presenting our findings.
>
> **Attention to Minor Details**
>
> All minor points raised in your review have been meticulously addressed and corrected. We believe these revisions significantly enhance the clarity and accuracy of our manuscript.
>
> We believe that our responses and revisions comprehensively address your concerns and contribute to the refinement of our paper. We are sincerely thankful for the opportunity to improve our manuscript based on your constructive feedback and eagerly await any further guidance you might offer.
>
>
>
> [1] Jia, Menglin, et al. "Visual prompt tuning." ECCV, 2022

---

> > ### Comment · Reviewer_HvDC · 2023-11-23
> > **Keeping my rating for acceptance**
> >
> > Thanks for your detailed response. I appreciate the efforts and clarification. I will keep my rating as accept.

---

### Official Review · Reviewer_hvFX · 2023-11-01

**Soundness:** 3 good
**Presentation:** 3 good
**Contribution:** 2 fair
**Rating:** 5
**Confidence:** 3

**Summary:**

This paper introduces visual prompt tuning (VPT) to improve proxy-based deep metric learning. As VPT could generate and integrate semantic proxies to improve the representations in deep metric learning, the proxies generated by VPT is considered to be better than random proxy which is generally used in proxy-based deep metric learning. The experiments are conducted several classification benchmark datasets including CUB-200-2011 (CUB200), CARS196, and retrieval benchmark datasets including Stanford Online Products (SOP), In-shop Clothes Retrieval (In-Shop). The proposed method is compared and shown to outperform or perform on par with various parameter-efficient fine-tuning methods such as Adapter Fine Tuning, and Bitfit.

**Strengths:**

Although the proposed method is simple, the proposed idea of introducing visual prompt tuning to improve proxy-based deep metric learning is well motivated and valid.

**Weaknesses:**

The main concern of this paper lies in the lack of evaluation/comparison:

-- The paper claims the proposed method to be parameter-efficient for deep metric learning (DML) tasks, but there is not comparison in term of efficiency as compared to existing DML methods.

-- The proposed method targets to solve deep metric learning (DML) tasks but the compared methods are mostly fine-tuning methods. More DML methods should be considered for comparison.

-- A baseline using default/vanilla proxy-based deep metric learning (without visual prompt learning) is missing to convince the effectiveness of introducing visual prompt tuning.

**Questions:**

What is the computational cost of the proposed method as compared to (1) vanilla proxy-based DML method, (2) other fine-tuning methods such as BitFit and Adapter? Please discuss the comparison on computation cost for model training and inference.

How does the proposed method compare to the state-of-the-art deep metric learning methods?

How does the proposed method compared to the vanilla proxy-based deep metric learning in terms of model performance and compute cost?

**Details Of Ethics Concerns:**

No concern on Ethics.

---

> ### Author Response · Authors · 2023-11-22
> **Gratitude for Your Insightful Review**
>
> Dear Reviewers,
>
> We extend our sincerest thanks for your insightful and constructive comments on our paper. Your feedback is invaluable in helping us clarify and address the questions you have raised.
>
> **Comparison in Terms of Efficiency with Existing DML Methods**
>
> Aligning with the metrics in existing parameter-efficient works [1][2], we assess efficiency by comparing the number of tunable parameters in the model, which reflects computational costs per iteration. Most existing DML methods [4][5] rely on full fine-tuning optimization with varying loss functions or sampling strategies, making their efficiency comparable to our full fine-tuning baseline.
>
> Furthermore, we have compared actual running latencies in Section C and Table 8 of our Appendix (formerly Supplementary Materials). This table shows the average running time per iteration in identical environments across 100 trials. Our results show that our proposed VPT and other parameter-efficient methods significantly reduce latency compared to full fine-tuning approaches. It's important to note that running latency is subject to hardware and software configurations. Hence, parameter-efficient works [1][2] often use tunable parameters as a universal metric for evaluating fine-tuning efficiency.
>
> **Consideration of More Existing DML Methods and a Vanilla Proxy-Based Baseline**
>
> In response to your concern, we have made the following clarifications and additions:
> 1) Our full fine-tuning approach, detailed in the main paper and Appendix, is based on the vanilla proxy-anchor (PA) [3] methodology, enhanced with a ViT backbone and larger ImageNet-21k pre-training datasets. We have included comparisons with state-of-the-art methods like Hyp-ViT [4] and RS@k [5], which also utilize a ViT backbone. These comparisons are specific to ViT backbones trained on ImageNet-21k, as earlier works [3][6][7][8] with CNN backbones and ImageNet-1k training show significantly lower DML benchmark performance.
> 2) Most current state-of-the-art methods, including Hyp-ViT [4] and RS@k [5], use full fine-tuning of the ViT with different losses or sampling strategies, similar to our PA baseline. These methods often require larger batch sizes (as indicated in Table 2), leading to higher latencies compared to our baseline. Therefore, their overall efficiency is close to or below our full fine-tuning PA baseline, while our VPT methods demonstrate higher DML performance, as shown in Table 2.
> 3) Although it's not entirely fair to compare with earlier works using different backbones and pre-training models, we have included performance and tunable parameters for some earlier works [6][7][8] and the vanilla proxy-anchor [3] as references. This comparison is also added to our Appendix as Table 6.
>
> ***
> |Method 	              | Arch | Parameters 	| R@1 | R@2| R@4| MAP@R |
> | :---------------------- | :------- | :----------------- | :------ | :------- | :----- | :------ |
> |ProxyNCA++[6]     | R50 | 26.5M 		| 69.0 | 79.8 | 87.3 | - |
> |NIR [7]	              | R50 | 26.5M             | 69.1  | 79.6 | -    | -  |
> |XBM[8]                  | BN	| 12M                | 65.8 | 75.9 | 84.0 |   - 	|
> |PA (vanilla)[3]       | R50 | 26.5M              | 69.7 | 80.0  | 87.0 | 26.5 |
> |Hyp[4] | ViT-S/16 | 30.2M			| 85.6 | 91.4 | **94.8** | - |
> |Full fine-tuning (PA) | ViT-S/16 | 30.2M  |  85.1  | 91.1 | 94.0 | 51.1 |
> |VPTSP (ours) | ViT-S/16 | **1.56M**          | **86.6** | **91.7** | 94.7 | **52.0** |
> ***

---

> ### Author Response · Authors · 2023-11-22
> **Gratitude for Your Insightful Review**
>
> **Q1: What is the Computational Cost of the Proposed Method**
>
> As mentioned, we evaluate our method's computational cost in two aspects:
> 1) The total trainable parameters that are updated in each optimization step (shown in Table 1).
>
> 2) The actual running latency per optimization step (shown in Table 8 of the Appendix).
>
> Table 1 in the main paper demonstrates that our method requires only about 5% of the trainable parameters needed for full fine-tuning methods. For running latency, our method significantly undercuts full fine-tuning methods, as listed in Table 8 of the Appendix.
>
>
> **Q2 and Q3: Comparison to State-of-the-Art DML Methods and Vanilla Proxy-Based DML**
>
> To better represent our "full fine-tuning method" as a stronger baseline compared to the vanilla proxy-anchor, we have added explanatory text in our revision. We have also updated Table 1 in the main paper and Table 8 in the Appendix for greater clarity. Additionally, a new table (Table 6 in Appendix) comparing our proposed method with existing DML works using ViT backbones [4][5] and earlier DML methods with CNN backbones [3][6][7][8] has been added.
>
> We hope these revisions and additional explanations comprehensively address your concerns, thereby enhancing the clarity and impact of our work. We are deeply appreciative of the opportunity to refine our manuscript based on your thoughtful feedback and eagerly await any further suggestions you may have.
>
>
> [1] Jia, Menglin, et al. "Visual prompt tuning." ECCV, 2022
>
> [2] Chen et al. "Adaptformer: Adapting vision transformers for scalable visual recognition." NeurIPS 2022
>
> [3] Kim, Sungyeon, et al. "Proxy anchor loss for deep metric learning." CVPR 2020.
>
> [4] Ermolov, Aleksandr, et al. "Hyperbolic vision transformers: Combining improvements in metric learning." CVPR 2022.
>
> [5] Patel, Yash, Giorgos Tolias, and Jiří Matas. "Recall@ k surrogate loss with large batches and similarity mixup." CVPR. 2022.
>
> [6] Teh et al. "Proxynca++: Revisiting and revitalizing proxy neighborhood component analysis." ECCV 2020
>
> [7] Roth et al. "Non-isotropy regularization for proxy-based deep metric learning." CVPR 2022
>
> [8] Wang et al. "Cross-batch memory for embedding learning.", CVPR 2020

---

### Author Response · Authors · 2023-11-22
**Summary of Reviewers' Comments and Revisions**

Dear Esteemed Reviewers,

We express our deepest gratitude for the valuable time and effort you have devoted to reviewing our manuscript. We are heartened by your recognition of:

1) The well-motivated idea (**hvFX**) and its novelty (**HvDC**,**1Fvy**).

2) The clarity and comprehensibility of our writing (**HvDC**, **1Fvy**).

3) Our proposed framework's improvement in DML tasks (**1Fvy**).

4) The well-organized experiments and their potential interest to the community (**HvDC**).

To address your concerns regarding:

1) The lack of comparison with state-of-the-art DML methods in terms of efficiency (**hvFX**,**1Fvy**).

2) There is a need for a comparison in terms of efficiency (**hvFX**).

3) The suggestion to include results from Appendix in the main manuscript (**HvDC**)

4) The suggestion to add analysis to the limited results(**HvDC**).

We have implemented several updates in our latest revision to enhance the manuscript's quality:

**Improved Accessibility of Supplementary Material:**

We have relocated the Appendix (formerly Supplementary Material) to directly follow the main paper, ensuring more accessible and more convenient access for readers.

**Clarification of Baseline Methodology:**

We have expanded our explanation to clarify that our full fine-tuning baseline adopts the proxy-anchor method, a state-of-the-art approach, as detailed in Table 1.

Furthermore, we elucidate that most existing DML methods, like our full fine-tuning baseline, incur comparable computational costs, as discussed in Section 5.2.

**New Comparative Table in the Appendix:**

A new table (Table 6), now included in the Appendix, concludes the efficiency of our methods with other existing DML methods.

**Enhanced Analysis of Limited Results:**

We have added a concise analysis of the limited performance observed in some benchmarks relative to state-of-the-art methods in Section 5.2. Additionally, we have incorporated an analysis of results on smaller pre-training datasets in Appendix Section B.2.

**Reorganization of Experimental Results:**

We have shifted our results on iNaturalist to the main manuscript and relocated the ablation study figure to the Appendix.

**Corrections and Updates:**

We have incorporated references to recent DML works from CVPR 2023, ensuring our manuscript aligns with the latest advancements in the field. We have also corrected typographical and grammatical errors in our revision.

We believe these revisions sufficiently address your concerns and suggestions. We remain dedicated to the ongoing refinement of our manuscript and look forward to any additional feedback you may offer.

We are profoundly thankful for the opportunity to improve our work based on your invaluable insights and anticipate the possibility of contributing to the research community through this publication.


Warm Regards,

Authors

---

### Meta-Review · Area_Chair_HbXk · 2023-12-05

**Metareview:**

This work aims to improve proxy-based DML by initialising proxy embeddings with semantic information from learnable prompts trained to correspond to class-specific features. The paper received diverging ratings which remain after the discussion periods. All reviewers agree that the proposed method is somewhat novel. There were consistent concerns around (i) missing comparisons to related DML methods; (ii) insufficient evaluation of the claim of parameter efficiency; and (iii) missing proxy-based deep metric learning baseline. The majority of reviewers found that their concerns were sufficiently addressed during discussion and recommend acceptance. The AC shares the majority opinion.

**Justification For Why Not Higher Score:**

Novelty is fine but not high and the problem is somewhat niche.

**Justification For Why Not Lower Score:**

The paper is well-written, the experiments are convincing, and the novelty is fine.

---

### Decision · Program_Chairs · 2024-01-16

Accept (poster)